# Optimizing Large Language Models with Automatic Speech Recognition for Medication Corpus in Low-Resource Healthcare Settings.

## Abstract

Automatic Speech Recognition (ASR) systems, while effective in general contexts, often face challenges in low-resource settings, especially in specialized domains such as healthcare. This study investigates the integration of Large Language Models (LLMs) with ASR systems to improve transcription accuracy in such environments. Focusing on medication-related conversations in healthcare, we fine-tuned the Whisper-Large ASR model on a custom dataset, Pharma-Speak, and applied the LLaMA 3 model for second-pass rescoring to correct ASR output errors. To achieve efficient fine-tuning without altering the full LLM parameters, we employed Low-Rank Adaptation (LoRA), which enables re-ranking of the ASR's N-best hypotheses while retaining the LLM's original knowledge. Our results demonstrate a significant reduction in Word Error Rate (WER) across multiple epochs, validating the effectiveness of the LLM-based rescoring method. The integration of LLMs in this framework shows potential for overcoming the limitations posed by conventional ASR models in low-resource settings. While computational constraints and the inherent strength of Whisper-Large presented some limitations, our approach lays the groundwork for further exploration of domain-specific ASR enhancements using LLMs, particularly in healthcare applications.

## 1 Introduction

Speech self-supervised learning has garnered significant interest owing to its encouraging results in several downstream tasks, and it has emerged as a novel tool for low-resource language speech recognition (Zhao & Zhang, 2022). Several studies have also used different speech models for downstream task on languages especially in low resource settings to achieve excellent results (Krishna et al., 2021). While this is good, a number of authors have purported in their study that Automatic Speech Recognition (ASR) models have shown good performance with English because it has been extensively tested but not on other low-resource languages (Yi et al., 2020). Even though this is true, we also find the supposedly better English performing models struggling with terminologies within the healthcare space.

Olatunji et al. (2023) in their study noted that the recent years have witnessed notable progress in the recognition of accented speech, as state-of-the-art (SOTA) automated speech recognition (ASR) models have become adept in transcribing a wide range of linguistic interactions But there is still a problem with these models' applicability in clinical or medical settings1, where nuanced communication is crucial and this is especially noticeable when physicians who don't speak English as their first language use ASR technology to record important medical data. Despite achieving low word error rates (WER) on speech in general, these SOTA models frequently have trouble reliably transcribing clinical named entities (NE) (Afonja et al., 2024). Furthermore, the majority of ASR systems are not tailored to the particular requirements of resource-constrained situations, where multilingual or noisy environment are widespread and computational resources may be inadequate. ASR tools have therefore been noted to not be optimal in the clinical settings especially in low resource settings due to wide accent variety, really noisy environment, multiple speakers at a time, as well as the frequent use of abbreviations within the medical conversation and minute errors in crucial information such as medication names, diagnosis, test findings, and lesion measures (e.g., writing hyper- instead of hypo- or rifampin instead of rifampicin) may jeopardize patient safety and put

medical professionals at unnecessary risk of lawsuits (Ajami, 2016). Hence, we decided to improve on one of the limitations of the ASR which is on drug names recognition.

Our study offers a unique and novel perspective to solving this problem. Rather than continue to good but didactic work of having to collect much more representative speech data from wide variety of speakers with different accent and intonation, we instead allow the ASR to transcribes as much as they can and then use Large Language Models (LLMs) for a second-pass rescoring method and to the best of our knowledge, this is the first of its kind done within the medication name domain although this has been extensively tested in other domains. (See Figure 1)

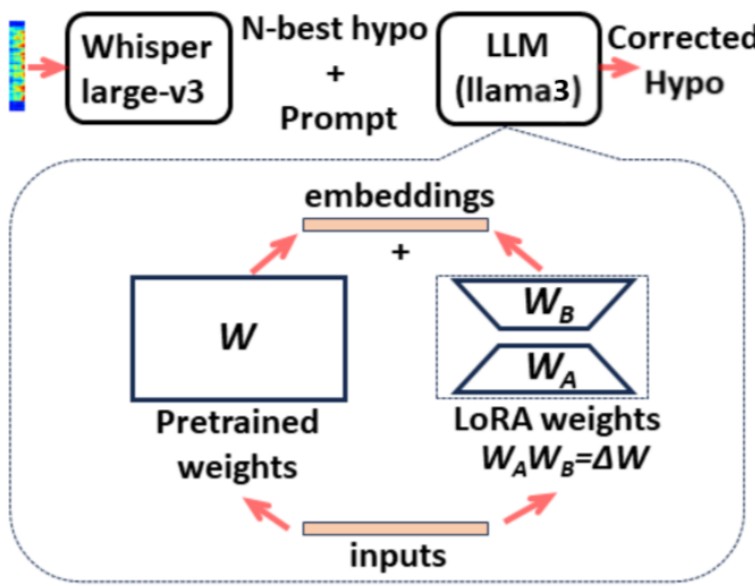

Figure 1: Workflow Diagram

## 2 RELATED WORKS

Chen et al. (2024) in their study proposed Hyporadise, a baseline for generative speech recognition with large language models. In another study, Li & Li (2023) combined LLMs with ASR for correction of Taiwan-Accentuated Text from an ASR. Similar to this study, Radhakrishnan et al. (2023) proposed Whispering LLaMa: A cross-modal fusion method intended for automated speech recognition (ASR) generative error correction. To produce correct speech transcription contexts, they made use of a system that makes use of both external linguistic representations and sonic information.

Efforts to integrate ASR and LLMs specifically for healthcare have been largely limited to high-resource settings, focusing on English-speaking populations and well-curated datasets. Works such as Kanithi et al. (2024) demonstrated the utility of LLMs for healthcare conversation and also proposed MEDIC; an evaluation method, but this assumes the availability of high-quality training data. Our research differentiates itself by focusing on optimizing ASR and LLM systems specifically for low-resource healthcare settings, targeting both domain-specific adaptation and the practical challenges of these environments, such as background noise and linguistic variability.

## 3 METHODOLOGY

This work employs LLM for error correction, which is illustrated in Figure 1 and involves second-pass rescoring in the output transcriptions produced by the ASR system (N-best decoding hypotheses). By inserting a neural module with a few extra trainable parameters to approximate the full

Table 1: Evaluation of the LLM Based Model

| Epoch | Result |
|-------|--------|
| 7     | 13.45  |
| 9     | 25.10  |
| 11    | 7.98   |
| 13    | 7.45   |

parameter updates, we introduce LoRA (Hu et al., 2021) to avoid having to tune the entire set of parameters of a pre-trained model. This allows for efficient learning of the N-best to transcription mapping without affecting the pre-trained parameters of the LLM. By adding trainable low-rank decomposition matrices to the current layers of LLMs, our approach allows the model to adjust to new data while maintaining the original LLMs fixed to preserve the prior knowledge. By injecting low-rank decomposition matrices 1, LoRA specifically executes a reparameterization of each model layer expressed as a matrix multiplication. The representations produced by the LLM are therefore not warped by task-specific tailoring. At the same time, the adaptor module gets the capability to forecast the real transcription from the N-best theories.

# 4  EXPERIMENT

## 4.1  EXPERIMENTAL SETTINGS

The experimental settings for fine-tuning a large language model in a low-resource environment are as follows.

1. LLM used: The experiment employs the Llama-2-8b Instruct model. This model is an instruct model good for chat completion and text generation.

2. ASR model: Whisper-Large-v3 generates 10-best outputs

3.The training is performed on Google Colab with an NVIDIA Tesla V100 GPU using 8-bit training. The hyperparameters for finetuning are 15 epochs, learning rate 1e-4, batch size 64, and LoRA rank $r = 4$.

4.Dataset: We used an open source dataset which had about 600 medication names prescribed globally with their trade names which we curated ourselves. It was separated to about 506 rows for the training and the rest for testing.

5.Evaluations: We used ROUGE score to evaluate the performance of the model

## 4.2  EXPERIMENTAL RESULTS

Table 1 shows the results of the experiment based on finetuning the LLM model. This result is significantly better than the finetuning of the ASR model itself with the use of speech dataset acheievng a benchmark of 21%.

## 4.3  LIMITATIONS

This study had a couple of limitation. The first being resource constraint thereby preventing us from being able to use the latest LLamA model and also GPU constraints which could have enabled us to run more inference to get optimal results. In addition, the list of dataset used in not holistic, as there are numerous other drugs that could not be captured. One important aspect realised is that some drugs are more pronounced as their chemical names rather than the brand names which was ysed in the dataset.

### 4.4 FURTHER DISCUSSIONS

More work can be done on the use of latest LLaMa or even more sophiscticated models. It would also be of interest to consider open source LLM that are domain-specific to the healthcare for example BioBERT, MEDITRON model and a host of others to the compare it with SOTA models like GPT 4 and LLaMa 3.1.

## 5 CONCLUSION

This paper demonstrates that combining a Large Language Model with a speech recognition system significantly enhances the recognition of medication names, even in low-resource environments. In the future, we plan to conduct further experiments to validate the method's effectiveness across a wider variety of scenarios.

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

## A APPENDIX

Code and dataset are available upon request

