# OpenReview forum: "Optimizing Large Language Models with Automatic Speech Recognition for Medication Corpus in Low-Resource Healthcare Settings."
_ICLR.cc/2025/Conference — Submitted to ICLR 2025_

### Official Review · Reviewer_PMeh · 2024-11-02

**Soundness:** 1
**Presentation:** 1
**Contribution:** 2
**Rating:** 1
**Confidence:** 4

**Summary:**

This work presents a brief experiment of applying LLMs to enhance ASR applied to the healthcare domain. The experiment applies LoRa as a finetuning of an LLM as a post processing of the transcription generated by a SOTA ASR like Whisper.

**Strengths:**

The idea is an simple and efficient way to build a system for a specific use case and potentially accelerate and decrease the cost and availability of automatic speech recognition tools on a sector as important for society as helathcare.

**Weaknesses:**

The document is brief and lack an exhaustive survey in the ASR space including other alternatives as finetuning LLMs for sumarization, rephrasing, plus alternative techniques to LoRA.

The domain and database is not described in detailed and the discussion and comparison with other techniques is not present. The statements are bold but they even don't demonstrate across multiple languages.

I believe authors need to dedicate more effort in improving the document  and make it more reproducible.

**Questions:**

a) Why you don't present an extensive survey on ASR?
b) Why you don't present ablation studies of you implement LoRA, amount of weights/layers  you alter etc..?
c) Why you don't describe the database more in detail and break down more results?
d) How this technique scales across multiple languages and the results and emerging properties of LLMs are transferred across languages?

---

### Official Review · Reviewer_HBSB · 2024-11-03

**Soundness:** 2
**Presentation:** 1
**Contribution:** 1
**Rating:** 5
**Confidence:** 4

**Summary:**

The paper describes adapting medical domain words using LoRA in LLMs so errors for ASR hypothesis can be corrected incase of low-resource settings. It showed experimental results where the accuracy improves after n-best rescoring with LoRA fine-tuned model

**Strengths:**

The paper describes the mechanism of adapting Medical names in LLMs so errors from ASR can be corrected

**Weaknesses:**

The paper itself has meagre details provided. Several limitations listed w.r.t data used by other papers needing high quality data, but the results presented in paper are also limited - dataset used in not holistic, as there are numerous other drugs that could not be captured, the chemical names are used instead of English words. There are no benchmarking evaluation and comparative study.

The paper doesn't talk about several other references which have used LLM for rescoring of ASR n-best hypothesis and how its approach is different

There is no visibly novel aspect in terms of approach except the domain in which this is applied seems under explored


Given the fact that there is no novel aspect reported in paper and insufficient benchmark, updated my review.

**Questions:**

below are some papers with extensive word done in LLM rescoring for ASR hypothesis. Please explain the novelty aspect in the paper. Also, please provide additional metrics with other LLMs, holistic data on a benchmark tests. Please provide comparative study


https://www.isca-archive.org/interspeech_2024/li24h_interspeech.pdf
https://arxiv.org/pdf/2406.18972

---

### Official Review · Reviewer_Fzh6 · 2024-11-03

**Soundness:** 1
**Presentation:** 1
**Contribution:** 1
**Rating:** 1
**Confidence:** 5

**Summary:**

The authors used Low-Rank Adaptation (LoRA) to fine-tune Whisper-Large ASR model on a custom dataset and applied the LLaMA 3 model for second-pass rescoring.

**Strengths:**

The authors described their experiments clearly.

**Weaknesses:**

The submission lacks originality and provides no basis for broader conclusions. The experiments conducted are simplistic and merely involve applying existing models to a customized dataset.

**Questions:**

What are the innovative aspects of this submission? What general insights can be gleaned from the experiments and discussions presented?

---

### Official Review · Reviewer_SVfL · 2024-11-04

**Soundness:** 1
**Presentation:** 1
**Contribution:** 1
**Rating:** 1
**Confidence:** 5

**Summary:**

The article proposes integration of automatic speech recognition systems with large language models to improve transcription accuracy of low-resource healthcare-focused medical transcription. The paper argues that employing low rank adaptation through enabling re-ranking of N-best hypothesis from whisper-based acoustic model, while retaining the LLM parameters, it is possible to improve transcription generation performance for low-resource medical transcription task.
While the proposed approach and the technical problem is quite relevant, but the paper significantly lacks content. I wonder if the authors were not able to complete drafting of the paper on time and hence submitted an incomplete paper. Many of the sections in the paper are incomplete, the most relevant sections did not have sufficient detail that shares what was done, what data was used, what analysis was performed and whether the argument made by the authors were indeed validated through the results form their experiments. The ratings shared in this review are predominantly based on the incompleteness of the article.
More detailed comments below: (1) Sections 3, 4 and 5 needs more content.
* For example in section 4 it is not clear how much data was used to learn the N-best to transcription mapping.
* Section 4 does not provide any information about the dataset used. It will be helpful to add a data description section and clearly outlining how much data was used for training, validation and evaluation purposes. How many speakers were there in the dataset? Were there speaker overlaps between the train-valid-test sets?
* It is also not clear if there was a baseline system that the proposed approach was compared against. Comparing against a baseline system usually helps to demonstrate the efficacy of the proposed approach.
* It is also not evident what the is the novelty in technology or algorithm that the article demonstrates.
* Conclusion (section 5) should provide a summary of the observations, and lessons learned from the investigation presented in the paper. Also providing some future directions, or rooms for improvement is always helpful.

**Strengths:**

The paper presents a relevant use case and a potentially viable technical solution to the challenges that one may face in such use case. However, the arguments posed in the paper were not sufficiently validated through experimental results and/or analysis.

**Weaknesses:**

The paper is largely incomplete. Sections 4 and 5 only contain placeholder texts with no detailed information about the data used, the modeling investigation performed, the results obtained and the analysis of the findings from the experiments. The paper also contain several typographic errors and strongly urge the authors to proof read the paper.
(1) Section 3, can benefit from having come detailed content on the dataset that was used. Given the low-resource nature of the target domain, it will be useful to specify how the data was collected, is the data publicly available? Providing some analysis of the data will also be useful, such as how much data was used (possibly in hours), some statistics on the number of speakers, gender, named-entities.
(2) Some insight on how the modeling was performed is highly desired, for example what was the train-valid-test splits. What metrics were used to evaluate model performance. How were the model parameters tuned.
(3) Section 4 does not specify any baseline system. Without a proper baseline system it is quite difficult to assess the merit of the proposed system. Comparing the proposed system against a baseline system usually helps to ascertain how useful the proposed system is, and whether the results presented significant.
(4) Conclusion in section 5 should share the lessons learned from the investigation presented in the paper, providing some future directions, or room for improvement.
(5) Typing errors: (a) section 1, page 1, line 044: "medical settings1" >> not clear what the 1 stands for, is it suppose to be a reference? (b) section 3, page 3, line 123, "low-rank decomposition matrices 1" >> not clear what the 1 stands for here as well.

**Questions:**

(1) It is not clear how big the dataset was, how many hours? how many speakers? what are the complexities of this dataset? Any prior art reported on this dataset? How was the data collected? Providing some analysis of the data will be helpful: (a) how much data was used (possibly in hours), (b) how many speakers, gender, named-entities. (c) how was the data split into train-validation-test sets? (d) does the test set had any speaker overlap with the train or the validation sets?
(2) The experimental section does not provide sufficient details. What evaluations were performed,? How much training data was used? Some insight on how the modeling was performed is highly desired, for example what was the train-valid-test splits. What metrics were used to evaluate model performance. How were the model parameters tuned.
(3) What was the baseline? Section 4 does not specify any baseline system. Without a proper baseline system it is quite difficult to assess the merit of the proposed system. Comparing the proposed system against a baseline system usually helps to ascertain how useful the proposed system is, and whether the results presented significant.

---

### Meta-Review · Area_Chair_7aR5 · 2024-12-14

**Metareview:**

This paper works on improving ASR accuracy on medication corpus with the technologies such as fine-tuned Whisper and 2nd pass rescoring with LLM which is LORA fine-tuned. These technologies are standard in the field. This paper serves as a brief experimental report using existing technologies with no clear novelty.

The paper submitted appears to be incomplete, consisting of only approximately three pages of content. The methodology section is limited to one paragraph, which briefly describes existing technologies. Section 4 offers only a brief overview of the model and dataset, lacking comprehensive detail. Additionally, the experimental results are presented in just three lines, with no accompanying discussion. There is no rebuttal during the authors/reviewers discussion. The authors are suggested to submit complete works next time.

**Additional Comments On Reviewer Discussion:**

There is no reviewer discussion.

---

### Decision · Program_Chairs · 2025-01-22

Reject